# Exploring the Determinants of Polypharmacy Prescribing and Dispensing Behaviors in Primary Care for the Elderly—Qualitative Study

**DOI:** 10.3390/ijerph20021389

**Published:** 2023-01-12

**Authors:** Najwa Taghy, Viviane Ramel, Ana Rivadeneyra, Florence Carrouel, Linda Cambon, Claude Dussart

**Affiliations:** 1Laboratory “Health, Systemic, Process” (P2S), UR4129, University Claude Bernard Lyon 1, University of Lyon, 69008 Lyon, France; 2U1219 Inserm Center, Bordeaux Population Health, University of Bordeaux, 33000 Bordeaux, France; 3ISPED, Bordeaux Population Health, University of Bordeaux, 33000 Bordeaux, France; 4CHU Bordeaux, 33000 Bordeaux, France; 5Central Pharmacy, Hospices Civiles de Lyon, 69002 Lyon, France

**Keywords:** polypharmacy, aging, qualitative study, intervention, prescribing, dispensing, theoretical domains framework, behavior change

## Abstract

Polypharmacy is particularly prevalent in the elderly. The interest in this issue is growing, and many interventions exist to improve the appropriate use of polypharmacy for older people. However, evidence of their effectiveness is still limited. Thus, the aim of this study, based on a qualitative approach, was to identify the key elements perceived to influence the prescribing and dispensing of appropriate polypharmacy to older people in primary care. Semistructured interviews were conducted with general practitioners and community pharmacists practicing in the region of Nouvelle-Aquitaine (France). Pre-existing topic guides based on the 12 TDF domains have been adapted and used. Data were analyzed using the framework method and content analysis. A focus group of healthcare professionals was conducted, and behavior change techniques (BCTs) were used to select the intervention components. Seventeen interviews were convened. A wide range of determinants were identified as barriers and/or facilitators. Nine domains were selected as key domains to target for intervention. Five intervention components (behavior change techniques—BCTs) to include in an intervention were finally selected. The results of this study will serve as a starting point for the design of a theory-based intervention targeting healthcare professionals to improve appropriate prescribing and dispensing of polypharmacy for older people in primary care.

## 1. Introduction

The world’s population is inexorably aging. An increase in life expectancy together with baby boomers entering old age have, among other reasons, accelerated population aging in most developed countries over the last ten years. The number of elderly people (65 years and older) in the world is predicted to double by 2050, going from 700 million in 2020 to 1.5 billion [1,2]. France is no exception. The elderly now represent 20.5% of the population, and the proportion of elderly in the population has increased by 4.7% in the last twenty years. In France, there are three more elderly people every five minutes [3]. This profound demographic shift has epidemiological implications, as the elderly are often polypathological. 

Polypathology is relatively well documented in the general population. This phenomenon has become the norm for people over 70 over the past twenty years [4,5]. According to a study published in 1997 by the “Institut de Recherche et de Documentation en Economie de Santé” (Institute of Research and Documentation in Health Economics, IRDES), 93% of people in this age group suffer from polypathology. The frequency of treated diseases was also high, with 86% being treated for at least two diseases in the past 12 months and almost half being treated for at least five [6]. The concurrent use of multiple medications is also becoming increasingly common [7]. Polypharmacy is generally defined, in patients older than 65 years, as taking five or more prescription medications [8]. However, no consensus concerning this definition exists (ref. [9]). 

Although often legitimate, polypharmacy can be problematic. Patients are at increased risk of iatrogenesis, compromising their quality of care by decreasing treatment compliance and increasing the risk of potentially inappropriate prescriptions. Polypharmacy has an economic cost to society in terms of health expenditure [10].

Interest in this issue is growing. Many interventions exist to improve the appropriate use of polypharmacy for older people. The literature provides numerous examples of interventions and programs to improve the quality and safety of prescribing, dispensing, and using medications for the elderly. These interventions target different populations depending on their objectives and concern the care professionals, the patients themselves, their caregivers, and their environment. However, the issue of the effectiveness of these interventions and their impact is at stake. Several studies have attempted to document and evaluate the effect of polypharmacy interventions; some also document the adverse events that may occur as a result of these interventions [11,12,13,14,15,16]. A very recent review of reviews highlights an association between polypharmacy interventions and reductions in potentially inappropriate prescribing behavior as well as improvements in medication adherence. However, evidence of their effectiveness for clinical and intermediate outcomes is still limited [17]. 

In relation to this issue, the Bilan partagé de medication (shared medication report, BPM) has been implemented in France since 2018. Conducted by a community pharmacist in coordination with a general practitioner (GP), it consists of a structured critical analysis of the patient’s medications with the objective of establishing a consensus with the patient regarding his or her treatment [18,19]. Indeed, both the GPs and the community pharmacists as primary healthcare providers play a pivotal role in mitigating this growing phenomenon. This shared report has truly formalized their shared follow-up of the elderly [20,21,22,23]. 

With this concern, we aimed to develop an evidence-based intervention to reduce the risks associated with inappropriate polypharmacy using a systematic approach and involving these key stakeholders. More precisely, our study investigated the behavior of GPs and pharmacists and the types of approaches that could improve the prescribing and dispensing of medication to the elderly requiring more than one medication. Based on a qualitative approach, the specific aim was to identify the key elements perceived to influence the behaviors of prescribing and dispensing appropriate polypharmacy to older people in primary care. A better understanding of these aspects is the first step in developing an intervention targeting the main determinants of inappropriate polypharmacy that is likely to reduce the associated risks. 

This study followed the UK Medical Research Council’s (MRC) framework for complex interventions. As part of initial intervention development work, the MRC recommends that researchers identify existing evidence and establish the intervention’s theoretical basis to overcome the limitations previously identified [24,25,26,27]. The study reported here used the theoretical domains framework (TDF) as the underpinning model of the theoretical determinants of behavior [28]. The TDF acts as a theoretical lens through which key determinants (i.e., theoretical domains) of the target behavior (i.e., prescribing and dispensing) can be identified for targeting with a behavior change intervention [29]. Key theoretical domains can then be mapped to appropriate behavior change techniques (BCTs) [30,31]. The selected BCTs form the “active ingredients” of the intervention and are used to bring about the required changes in the target behavior [32]. This approach offers a robust, systematic, and theory-based approach to selecting and specifying components of a complex behavior change intervention [33]. 

The objectives of this study were to (i) identify the determinants (i.e., barriers and/or facilitators) of the behaviors of prescribing and dispensing appropriate polypharmacy to older people in primary care; (ii) select key TDF domains to target to achieve desired changes; and (iii) map key domains to appropriate BCTs (intervention components) and select those to include in an intervention that could feasibly be delivered by GPs and community pharmacists.

## 2. Methods

A study protocol was designed to conduct a qualitative study among GPs and pharmacists in the Nouvelle Aquitaine region in France [34]. Ethical approval was obtained from the Collège universitaire de médecine générale (University College of General Medicine) on 23 July 2020 (n° IRB: 2 July 2020).

Semistructured interviews were conducted with GPs and community pharmacists using adapted TDF-based topic guides to identify the determinants (i.e., barriers and/or facilitators) of the behaviors of prescribing and dispensing appropriate polypharmacy to older adults and to select the key areas of the TDF to target to achieve the desired changes. These selected domains were subsequently mapped, using two different reference sources, to identify BCTs that would be the components of an intervention to improve appropriate polypharmacy in older patients. Finally, a focus group composed of GPs, a geriatrician, an addictologist, and a pharmacist was convened to consolidate the consensual discussion of the research team in the final step of selecting BCTs to include in the intervention. Figure 1 illustrates how different qualitative methods were used to respond to the study’s three main objectives.

### 2.1. Semistructured Interviews

#### 2.1.1. Constitution of the Panel

To obtain a representation of GPs’ and pharmacists’ perspectives, meanings, opinions, and ideas about polypharmacy and to capture the variability of discourses on this topic, we targeted individuals with relevant characteristics. These characteristics included intentionality and relevance to prescribing and dispensing medications to older adults in primary care [35]. The main selection criterion was the geographical location of the GP practice and pharmacies in which the GPs and pharmacists practiced in order to target the different geographical areas with a large population of elderly people (60 years and older).

With 29% of people aged 60 or over (compared with the national average of 25%), Nouvelle Aquitaine is the oldest region in France. Almost all of these seniors (95%) live at their own home [36]. Based on the available data from The French National Institute of Statistics and Economic Studies (INSEE), seven areas in this region with an aging population—specifically, municipalities where the proportion of the population aged 60 years or more exceeded 40%—were identified (area 1 and 2 in Gironde canton, area 3 in Pyrénées-Atlantiques canton, area 4 in Landes canton, area 5 and 7 in Dordogne canton, and area 6 in Lot-et-Garonne canton).

The constitution of the panel of providers was conducted in two stages. GPs have been identified in the previously defined geographical areas using the Yellow Pages [37]. Lists were drawn up, and a random selection of doctors to contact was made. Contacted by telephone, the interviewer (NT) gave a brief overview of the study before inviting GPs to participate. An invitation letter and an information sheet with a summary of the project were sent to those who expressed an interest to receive additional information. In the context of the COVID-19 epidemic, the interviews were conducted exclusively by telephone. The selection of pharmacists was based on the network of recruited GPs. The recruited GPs were asked to identify the local pharmacies that dispense the majority of prescriptions for their elderly patients. Similarly, NT contacted pharmacists by telephone to arrange an interview with consenting pharmacists. Written informed consent was obtained from all participants before taking part in the research. All participants were financially compensated for their time (cheque for EUR 50). 

Initially, the estimated panel size was 14–20 GPs (ideally 2 and at least 1 from each of the seven zones) and 7–10 pharmacists (at least 1 from each of the seven zones), representing a panel of 21–30 participants. However, during the study, the semistructured interviews were analyzed, and the inclusion of participants was stopped when no new information or themes emerged from the data and when interview n + 1 did not provide anything new compared to interview n [38,39,40]. 

#### 2.1.2. Process of Semistructured Interviews 

Semidirective interviews were conducted by NT with recruited professionals by phone between November 2020 and December 2021. Pre-existing thematic guides validated by researchers’ consensus were used as a base. These included two guides based on the 12 domains of TDF, one for GPs (to study prescribing behavior), and another for pharmacists (to study dispensing behavior). Each topic guide included a series of similar questions covering four key areas: professionals’ views on the term ‘polypharmacy’; professionals’ assessment of a clinical scenario depicting an older patient receiving inappropriate polypharmacy; professionals’ perceptions of barriers and facilitators to ensuring the prescribing (GPs) and dispensing (community pharmacists) of appropriate polypharmacy to older people; and professionals’ views on potential intervention components and outcome measures for inclusion in future intervention studies. These interview guides were tested, validated, and used during the survey conducted by an Irish research team as part of a major multiphase research project conducted in 2015 [41,42]. In order to adapt them to the needs of the present survey in France, the guides were translated and adapted. Specifically, the wording of some questions was modified to better fit the context of the French health system. The first version of these two adapted interview guides was elaborated by NT and discussed with the fellow researchers. They were piloted beforehand during one-on-one interviews with two GPs and two pharmacists who were not included in the study sample. If required, questions were reworded, clarified, or completed in order to obtain final versions. 

The average duration of the interviews was 45 min. Interviews were audio-recorded in full using a voice recorder. After the interview, nonverbal aspects were noted, as well as general impressions of the interview process. A verbatim transcript was made to faithfully reflect the entire content of the interview. The quality of the transcription was checked for each recording. Recordings were separated into two groups to distinguish the interviews with GPs from those with pharmacists before being anonymized.

### 2.2. Focus Group 

A focus group was held in June 2022. It included two GPs, one GP-geriatrician, one GP-addictologist, and one pharmacist. Two of the GPs had taken part in the interviews. The focus group was organized via videoconference by NT who acted as moderator and notetaker. A two-hour session was scheduled. It started with a presentation to explain the context, define the concepts, and introduce the objective of having a consensus discussion around the selection of the intervention components. A table with the BCTs that had been mapped to the key domains was used as a support. In the second step, in order to move closer to a consensus, the table was sent back to the participants by e-mail three days later to collect their answers and comments. 

### 2.3. Data Analysis 

The strength of this method, which is at the basis of the definition of all intervention theories, is that it combines data from the scientific literature (literature review) on the one hand with data from practical and contextual knowledge (new qualitative data from practitioners and pharmacists) on the other. As well, our study protocol provides triangulation at several levels: (a) triangulation of data with two data collection techniques (interviews and focus group) and two data sources (interviews with GPs and pharmacists) and (b) triangulation of investigators with the association of several researchers for data analysis (double coding). These data from different sources were exploited in a three-step analysis as follows (Table 1):

#### 2.3.1. Identification of Determinants (Barriers and/or Facilitators) of Appropriate Polypharmacy-Prescribing and -Dispensing Behavior

There was an initial stage where the transcripts were read, proofread, and checked before being analyzed. A thematic analysis was conducted. NVivo software was used to facilitate the analysis of the data. The data were indexed and classified in a matrix following the framework method [43], and the TDF1 was used as the analytical framework and its 12 domains as the coding categories.

A two-stage analysis was adopted. Firstly, a deductive approach using predefined coding categories was used. NT initially coded all transcripts. At the end of the study, a random selection of transcripts was made to select a third of the cases, which were given to two other researchers (VR and AR) for coding. The coding was compared, and points of discrepancy were discussed by the three coders. The data were managed using NVivo QSR 10 by NT and conventional tables for the other two coders. The data were then imported to generate a framework matrix.

Secondly, using an inductive approach, a content analysis was carried out based on the previously obtained framework matrix, which allowed the emergence of themes relating to the determinants (barriers and facilitators) of the behaviors of prescribing and dispensing appropriate polypharmacy within each TDF domain. The results of the analysis were used as a basis for identifying key theoretical areas and selecting BCTs.

#### 2.3.2. Identification of Key TDF to Target

The decision of the relevance of each domain to the target behaviors (i.e., prescribing and dispensing appropriate polypharmacy) was justified by the extent to which sections of the interview transcripts were coded for each domain (whether the participants related the domain to the target behavior). This selection was based on the evidence from the literature review as well as the contributions of previous research, and it was validated through a consensus approach by the researchers involved in the study, taking into account the resource feasibility of the project and its objectives. 

#### 2.3.3. Identification and Selection of the Components of the Intervention

We have mobilized the mapping process and the method reported by the Irish team (Cadogan et al.) that conducted a similar study. We adapted the guidelines of the Irish team for the needs of our survey as mentioned above [41,42] and based on two reference sources: Michie’s original matrix [31] and the matrix produced by Cane’s team, where a table of BCTs has been reliably allocated to the 14-domain version of the TDF by a panel of behavior change experts [44]. A table was compiled from the list combining all BCTs that had been matched to the key domains in each reference source. This table was presented to the group of healthcare professionals (HCPs) in the focus group. The aim was to reach a consensus on which BCTs should be selected for inclusion in the intervention. Participants were asked to say whether they would include the BCT, to justify as far as possible if not, and to give an example of operationalization if they would. The same table was sent out by mail to collect as many opinions as possible. Then, the researchers analyzed all the available data (interviews, focus groups, and the literature), compared them, and pooled them in order to obtain a consensus on the BCTs. 

## 3. Results

### 3.1. Participants’ Characteristics

A total of 17 participants (11 GPs and 6 community pharmacists) were recruited. Despite repeated efforts, it was not possible to recruit a pharmacist in area 7 (Table 2). Data saturation was reached after eight interviews with GPs and four interviews with community pharmacists. The duration of interviews ranged from 37 to 85 min for GPs and 34 to 70 min for community pharmacists.

### 3.2. Summary of Key Findings from the Identification of Behavior Determinants Stages (Barriers and/or Facilitators)

The interviews identified a wide range of factors in each TDF domain that were perceived to influence appropriate polypharmacy prescribing and dispensing to older people. A table available as an additional file (Additional file 1) describes all the factors identified according to TDF domains and provides illustrative quotes. For example, concerning the TDF domain related to knowledge, four facilitators were identified: (i) knowledge acquired in initial or continuing training (“…I rely on initial training and continuing education especially…” (MG3)), (ii) consulting/receiving recommendations (“…yes (we receive recommendations) for example on antihypertensive drugs to be used in the elderly…” (MG6)), (iii) subscription to the journal “Prescrire” (“I subscribe to the journal Prescrire and I’m interested in it. Prescrire also criticizes all drugs…the only information I get from Prescrire…” (MG4)), and (iv) using databases and software (“I look at Vidal for the indication, the type of drug if I really don’t know it and for what indication it can be prescribed. After that, I sometimes look on the internet for therapeutic indications…” (MG2)). One element regarding the knowledge of geriatric therapeutics was also identified and classified as both a barrier (“It’s true that we don’t necessarily have a lot of training on the geriatric side knowing that geriatrics at the therapeutic level is particular…” (MG2)) and a facilitator (“…yes (we receive recommendations) for example on antihypertensive drugs to be used in the elderly etc…yes we try, yes. The benzo etc…try to give hypnotics or short half-lives, yes, we try.” (MG6)).

As the analysis progressed, it emerged that some determinants were relevant to several TDF domains at once. It seemed more relevant to present a table crossing the identified factors (barriers and/or facilitators) on one side and the TDF areas concerned in each one (Table 3).

### 3.3. Identification of Key Domains

The previous step identified a number of determinants in each of the 12 domains. All of these domains are important for the targeted behaviors of prescribing and dispensing appropriate polypharmacy to elderly patients. However, the examination of the determinants in each of the domains, whether they were identified as barriers and/or as facilitators, according to the importance of their representation in the interviewed GPs’ and pharmacists’ discourse, allowed us to select the domains likely to be more relevant to target in the context of the intervention to be developed in our project. Three TDF domains were excluded (knowledge, beliefs about consequences, and nature of the behaviors). The remaining nine domains (skills; social/professional role and identity; beliefs about capabilities; motivation and goals; memory, attention, and decision processes; environmental context and resources; social influences; emotion; and behavioral regulation) were selected as key domains to target for intervention.

### 3.4. Selecting the Components of the Intervention 

Based on the two reference sources [31,44], 49 BCTs corresponding to the selected key domains are listed (Table 4).

Based on focus group discussions completed by subsequent feedback from participants by e-mail, it was possible, in light of the interview data and considering all the parameters of interest, to select five BCTs to include in the intervention. This result can be represented as follows (Figure 2): 

## 4. Discussion

According to the most recent version of the Cochrane review designed to determine which interventions, alone or in combination, are effective in improving the appropriate use of polypharmacy and reducing medication-related problems in older people, published studies have failed to establish a clear link between interventions and clinically significant improvements. However, it highlighted that it is possible based on the interventions’ modestly beneficial impact in terms of reducing potential prescription omissions [12]. While these conclusions need to be put into perspective because they were based on a very limited number of studies, many of which had a risk of bias, they have been confirmed in many recent reviews. Polypharmacy interventions are associated with a reduction in potentially inappropriate prescribing (PIP) and medication adherence improvements [17]. Given that PIP is a multi-faceted problem involving multiple stakeholders, interventions addressing this problem require multiple components to target the different stakeholders’ behaviors [14]. We opted for an approach targeting the key actors’ behaviors, i.e., prescribing by GPs and dispensing by pharmacists, and for an intervention development modeled on the Medical Research Council (MRC)’s guide for the development and evaluation of complex interventions [27]. We also chose to follow the TDF-based method for the systematic identification of behavior change determinants. As the TDF method was originally developed to study the implementation of evidence-based practices by health professionals, it provides a theoretically sound basis for intervention design, allows explicit links to be made between intervention components and outcomes, and, ultimately, helps to understand the causal mechanisms underlying the intervention’s effects [28,45].

In practice, the data collected revealed a wide range of perceived barriers and facilitators to appropriate polypharmacy-prescribing and dispensing behavior (Appendix A). Our results at this stage were very similar to those found in the surveys carried out in 2016 by the French Directorate for Research, Studies, Evaluation and Statistics (DREES), which aimed to study the opinions and practices of French GPs with regard to the management of multimorbidity and to document their role towards their specialist or pharmacist colleagues and their strategies for managing polyprescriptions and deprescribing (ref. [46]). As with most of the GPs in our panel, a large majority of the GPs in these surveys assume their central role in managing the prescribing of these patients and feel comfortable suggesting deprescribing drugs they consider inappropriate. More than half of them felt that collaboration between GPs and pharmacists on polypharmacy management was insufficient, and this is also what the majority of professionals interviewed in our survey in Nouvelle Aquitaine deplored.

Based on these findings, the challenge for the researchers was to subsequently select the key domains to target and the intervention components to bring about the desired behavior change. Indeed, these factors should be considered as levers that act on mechanisms to facilitate or, on the contrary, as barriers to hamper the execution of the desired behavior. They can act on several mechanisms at the same time and even interact with each other.

For example, the “ patient profile”, depending on whether the patients are helping or opposing their treatment change, requires specific skills from healthcare professionals (“skills”), can generate anxiety in patients and stress in professionals (“emotion”), involves adopting strategies to explain, negotiate, and defer (“behavioral regulation”), and necessarily acts on the professionals’ motivation and the priority they give to putting energy into this effort to convince the patients to achieve the desired result (“motivation and goal”) *(“There are several profiles of patients; there are those who as soon as you remove something it is “ah! I miss the tablet so-and-so…etc.” and there you go for a big explanation”—GP1)*. Moreover, it is clear that this factor inevitably interacts with other factors such as the “lack of specific training on communication skills”, “lack of time and resources”, and “HCPs tiredness”, among others. There is, therefore, a continuum of behavioral change for doctors and pharmacists, and what is at stake is reaching a complementarity in mechanisms to activate. While following this reasoning, the researchers chose nine key domains and excluded the three domains that seemed to be relatively less relevant, according to the feedback from our survey.

Indeed, with regard to “knowledge”, several factors were mentioned by the interviewees as facilitators of their resource access necessary for appropriate prescription and dispensing. It requires initial and ongoing training, recommendations, and information via medical journals or support from prescription assistance software. Sometimes, the professionals complained about a lack of training or specific recommendations for the elderly’s therapeutic care (*“It’s true that we don’t necessarily have a lot of training on the geriatric side, knowing that geriatrics is very specific in terms of therapy”—GP2*), but from our point of view, acting on these levers would likely prove difficult.

On the other hand, we have chosen to exclude the field “Beliefs about consequences “ because, in all the interviews, the GPs and pharmacists were aware of the extent of the phenomenon of polypharmacy in France, which they all link, quite logically, and a bit like a fatality, to polypathology *(“…there is everything. Some have two or three. There are some who can have fifteen. It will depend on the pathologies. No, usully, doctors try to limit the number of medications as much as possible, but sometimes you can’t do otherwise anyway. On average seven or eight…’—PH3)*. They were all aware of the consequences of prescribing and/or delivering inappropriate polypharmacy to elderly patients. They had an unwavering belief in the importance of avoiding iatrogenic effects and other immediate and short-term consequences. Most of them also were concerned about managing the treatments as best as possible to avoid “domino effects” or rather serious consequences such as complications and/or hospitalization, impacting their patients’ long-term health.

The TDF domain “Nature of the behaviours” was also excluded because the question “Is there anything you do regularly in your daily practice to ensure that you are prescribing/dispensing appropriately to elderly patients?” aroused very little interest among the interviewees in our study and, again, the leeway on the levers identified seems very limited. From the GPs’ discourse, more than pharmacists for that matter, we perceive that they already engage in those behaviors but implicitly and as a part of their work routine, such as always checking their prescriptions before printing them, double checking the issued prescriptions, always keeping in mind the benefit/risk balance, knowing the recommendations, and even receiving help from software. On the other hand, they seem quite opposed to the idea of having self-imposed or external monitoring as this would imply changing their practice in the context of challenging working conditions. 

From the mapping of the selected domains with the BCTs, we obtained a table with 51 items. Selecting those most relevant to integrate into an experimental intervention to be developed was a new challenge. The choice to include or exclude each BCT was discussed among the focus group participants. This choice had to take into account all the acceptability and feasibility parameters, informed by the interview survey results and previous research. The objective was to ensure that the intervention would be feasible for health professionals in their daily practice conditions. Despite this, three BCTs (“graded tasks”, “behavioral rehearsal/practice”, and “self-monitoring”) that are likely to require repeated administration and/or extended time periods to affect required changes in target behaviors, according to previous studies [42], were selected by the focus group and had to be subsequently withdrawn by the researchers after consultation. Indeed, time turns out to be, unsurprisingly, a determining factor more generally impacting the healthcare professionals’ daily practice. In France, in the context of a scarcity of resources and medical supply, we can even deplore a real crisis. The DREES projections suggest a sharp decline (−24% compared to 2012) in the number of private sector doctors by 2027. This would be particularly marked among GPs (−30%). Combined with the French population growth and aging, these declines would result in the density of private practitioners weakening nationwide. In response to this, doctors are being forced to adapt their care practices by extending their working day or shortening the time spent on professional training. This has direct consequences for patient care: 54% of GPs say they have to increase the time taken to make appointments, 53% refuse new patients, 40% limit some patient follow-up, and 28% shorten the length of appointments [47].

Two BCTs, “Verbal persuasion to boost self-efficacy” and “Focus on past success”, were also highlighted by the focus group but were finally withdrawn after consultation and regarding the qualitative study results. In fact, in the interviewees’ discourse, the doubts regarding their ability to adopt the optimal behavior were essentially linked to the practice conditions, the lack of time, and issues accessing the patient’s medical records and other information that would enable them to have the necessary overview to prescribe and deliver with confidence. It does not seem relevant to help them to strengthen their self-efficacy nor to comfort them in relation to their past experience. 

It is interesting to see that the BCT “ Goal/target specified: behaviour or outcome”, which was discarded in a similar study by Cadogan et al. with the justification that, ideally, the target behavior should be adopted by all elderly patients, does not seem to shock the French health professionals at all. They feel that it could be encouraging to set a goal/target—not a quantitative one but dealing with elderly patients’ profiles with specific criteria based on which they would adopt the target behavior, for example. In the same field, another BCT (“Planning implementation/action planning”) was selected during the focus group. This involves asking healthcare professionals for detailed planning of the desired behavior execution [32]. This could consist of setting explicit inclusion criteria to target patients who are prescribed/distributed appropriate polypharmacy by healthcare professionals [42].

Similarly, in the Irish study, the BCT “ Environmental changes (e.g., objects to facilitate behavior)” was considered out of the project scope. Yet, in our study, both during the interviews and during the focus group, the professionals very strongly expressed the need for reorganizing their work environment to gain time by relying on the nurses, whose role is just as important as theirs according to them, using interns, or improving the working and communication conditions within the medical office, the pharmacy, or, more broadly, the formalized networks of healthcare professionals. Another study published by the DREES from the 2018–2019 general medicine practice and conditions observation panel echoes our results. It mentions that the use of digital health tools, such as electronic medical records, prescription assistance software, and secure health messaging, decreases according to general practitioners’ age. Indeed, nearly 80% of those under age 50 use these three tools daily compared with only 48% of doctors aged 60 or over. In addition to age, the use of these tools goes hand in hand with a more advanced collective organization. A total of 75% of doctors practicing with other general practitioners use the three tools compared to 58% of those who only practice general medicine [48]. These are very widely mentioned levers by the GPs interviewed, and their integration into the intervention seems imperative.

The BCT “Social processes of encouragement, pressure, support “ was also kept. Identified as an important element, it acts on different mechanisms involving four TDF domains (social influences; beliefs about capabilities; social/professional role and identity, and emotion). The focus group participants mentioned important leverage that can be provided by the involvement of a geriatrician or psychiatrist colleague, for example, to help the GP with complex cases. The Cadogan study envisaged an operationalization that would consist of pharmacists receiving a list of preapproved patients from the GP practice, which would encourage/support them in engaging with patients to ensure that they are dispensed appropriate polypharmacy [42]. In our study, where pharmacists were asked about their perception of their role in ensuring appropriate polypharmacy for elderly patients, this lack of support from GPs, in a context where the role of each is not clearly defined, seemed to be a significant barrier. This is certainly not unrelated to the fact that, in a survey of community pharmacists in the Auvergne-Rhône-Alpes region, 42.2% of the pharmacists identified the refusal of the GP to adhere, along with the lack of time, as the main barrier to the proper functioning of the BPM [49]. The BPM, set up in France in 2018, consists of interviewing the patient and performing a pharmaceutical analysis of their prescriptions. It is based on multiprofessional cooperation between pharmacists, GPs, and other healthcare professionals [18,19].

The BCT “Modelling/demonstration of behaviour by another” was not the subject of a consensus during the focus group but was, nevertheless, retained by the researchers after consultation. Most of the GPs and pharmacists said that they were concerned about keeping up to date with the latest recommendations through training, reviews, and digital resources. They also expressed the need to be trained specifically with regard to communicating with the elderly to have the resources to motivate them when they are faced with a reluctant or anxious patient. However, the lack of time appears to, once again, limit the amount of time devoted to training [47]. The speed of medical progress, exemplified by the staggering number of publications, results in the generation of numerous recommendations. It is increasingly challenging for professionals to absorb and apply these in their daily practice. Several factors, including the mechanism adopted for their dissemination, determine whether professionals adhere to recommendations. There are several dissemination strategies that can be applied in isolation or in combination. While simple dissemination methods (distribution of printed or audiovisual documents and continuing medical education) fail, there is promising evidence that presentations by “opinion leaders”—health professionals designated by his or her colleagues as influential in terms of training—can be effective in this respect [50]. A demonstration of how to prescribe or deliver appropriate polypharmacy by an “opinion leader” during a typical encounter or consultation with an elderly patient would allow health professionals to acquire these communication skills in a relatively short time.

### Limitations

The first limitation of this research is that our study was hampered by a difficult context. Indeed, the COVID-19 health crisis complicated a context in which general practitioners (GPs) and pharmacists were desperately short of time. This made the recruitment stage particularly difficult. A second very important limitation of our research is that we did not include nurses. From the design of the protocol, we chose to target GPs and pharmacists. However, from the very first interviews, the role of nurses in accompanying elderly patients in their outpatient care proved to be essential in support of that of GPs and pharmacists. Finally, in the analysis stage, we identified another limitation related to the study of the context and practice environment of GPs and pharmacists in the selected areas. These are, of course, elements that we collected during the interviews; however, it would have added more robustness to our analysis to have more detailed knowledge and factual elements concerning the contexts and conditions of practice in each zone.

## 5. Conclusions

Using an approach combining a systematic identification of the determinants of behavior change, using the TDF, and the selection of intervention components based on BCT, the data from this qualitative study based on interviews and a focus group allowed us to identify a number of levers, acting through different mechanisms, to achieve behavior change in GPs and community pharmacists. Our results will serve as a starting point for the design of a theory-based intervention targeting healthcare professionals in the Nouvelle Aquitaine region to improve appropriate prescribing and dispensing of polypharmacy to the elderly in primary care. 

## Figures and Tables

**Figure 1 ijerph-20-01389-f001:**
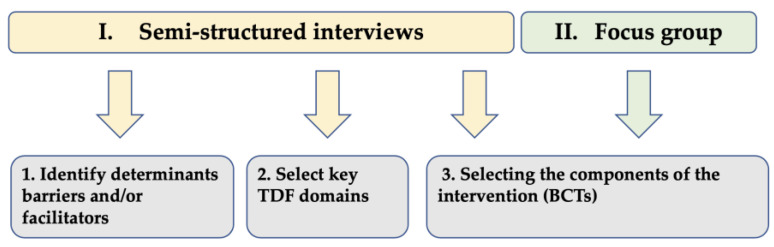
Steps of the qualitative study.

**Figure 2 ijerph-20-01389-f002:**
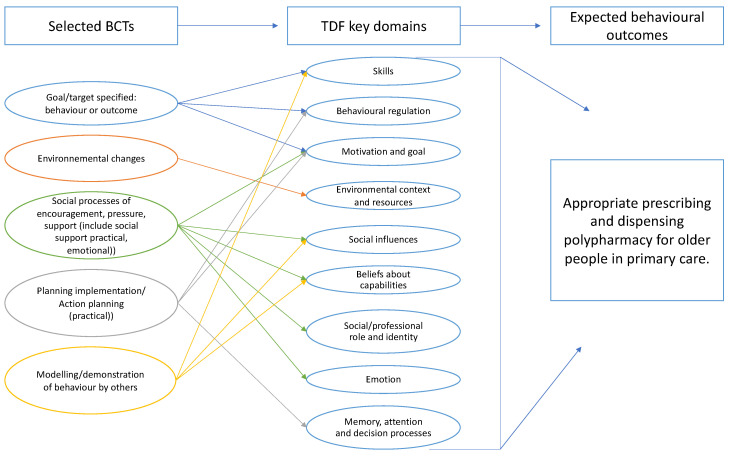
Selected behavior change techniques by key theoretical domains.

**Table 1 ijerph-20-01389-t001:** Origin of data by stage of analysis.

	Evidence from Literature	Semistructured Interviews	Researchers’ Consensus	Focus Group
1. Identify determinants: barriers and/or facilitators		X	X	
2. Select key TDF domains	X	X	X	
3. Select the components of the intervention (BCTs)	X	X	X	X

**Table 2 ijerph-20-01389-t002:** Participants’ characteristics.

	General Practitioners (n = 11)	Pharmacists (n = 6)
Gender Male Female		
	4	5
	7	1
Years of professional experience	2–33	1–25
Activity area		
Area 1	3	1
Area 2	2	1
Area 3	2	1
Area 4	1	1
Area 5	1	1
Area 6	1	1
Area 7	1	0

**Table 3 ijerph-20-01389-t003:** Identified determinants (i.e., barriers and/or facilitators) of general practitioners’ and community pharmacists’ behaviors. grey boxes simply indicate: factor identified as having an influence on the TDF domain.

	TFD Domains	Knowledge	Skills	Social/Professional Role and Identity	Beliefs About Capabilities	Beliefs about Consequences	Motivation and Goals	Memory, Attention and Decision Processes	Environmental Context and Resources	Social Influence	Emotion	Behavioral Regulation	Nature of the Behaviors
Determinants													
knowledge acquired in initial or continuing training													
knowledge of geriatric therapeutics													
Consulting/receiving recommendations													
keep informed (e.g., by subscription to the journal “Prescrire”)													
Using databases and software													
Patient profile: helping or opposing													
Lack of specific training on communication skills													
Trusting relationship GP, Pharmacist/Patient: relational aspect, interactions, explanations, shared decision													
More or less effective communication between prescribers, and between doctors and pharmacists													
Functioning in a formalized network of professionals													
GPs conscious or not of their pivotal/oversight role													
Pharmacists’ belief in their firewall role													
GPs’ belief in the complementary role of pharmacists													
Pharmacists’ confidence in GP prescribing													
GPs feeling comfortable about their role in relation to specialists													
lack of resources and time													
Complementary role of home care nurses													
Lack of formalization of the roles of each in the patient’s pathway													
Lack of access to the whole information in the patient file													
Constant concern to avoid iatrogenic: drug interactions and side effects													
Dealing with short- and long-term consequences													
Belief that polypharmacy goes hand in hand with polypathology													
Belief in the impact on drug compliance													
Benefit-risk balance													
HCPs Tiredness													
Patient condition													
High motivation to systematically check the medication													
Employing strategies: deferring, explaining, negotiating													
Being a training supervisor													
Experience													
Habit/ chronic patients/ prescription refills													
Reducing prescription renewal intervals													
Verification stage before printing the prescription by the GP-double check by the pharmacist when dispensing													
Sharing a practice with other doctors													
The feeling of being completely free in prescribing and dispensing													
Anti-lab generation of GPs													
Financial aspect for pharmacists													
Pressure from patient and/or family													
Patient profile: anxiety patients and psychiatric profiles													
HCPs’ confidence in their ability to cope with stress													
Relatively limited access to indicators and statistic													

**Table 4 ijerph-20-01389-t004:** Description of the TDF key domains selected according to the bibliographic reference used.

Domain Label	BCTs Identified from Reference [31]	BCTs Identified from Reference [44]
Skills	Graded tasks Behavioral rehearsal/practice Habit reversal Body changes Habit formation	6. Goal/target specified: behavior or outcome 7. Monitoring 8. Self-monitoring 9. Rewards; incentives 10. Graded task: start with easy tasks 11. Increasing skills: problem solving, decision making, goal setting 12. Rehearsal of relevant skills 13. Modeling/demonstration of behavior by others 14. Homework 15. Perform behavior in different settings
Social/professional role and identity	None	Social processes of encouragement, pressure, support
Beliefs about capabilities	Verbal persuasion to boost self-efficacy Focus on past success	3. Self-monitoring 4. Graded task: start with easy tasks 5. Increasing skills: problem solving, decision making, goal setting 6. Coping skills 7. Rehearsal of relevant skills 8. Social processes of encouragement, pressure, support 9. Modeling/demonstration of behavior by others 10. Homework 11. Perform behavior in different setting
Motivation and goals	Goal setting (outcome) Goal setting (behavior) Review of outcome goal(s) Review behavior goals Action planning (including implementation intentions)	6. Goal/target specified: behavior or outcome 7. Rewards 8. Graded task 9. Increasing skills 10. Social processes of encouragement, pressure, support 11. Persuasive communication 12. Information regarding behavior or outcome 13. Motivational interviewing
Memory, attention, and decision processes	None	Self-monitoring Planning, implementation Prompts, triggers, cues
Environmental context and resources	Restructuring the physical environment Discriminative (learned) cue Prompts/cues Restructuring the social environment Avoidance/changing exposure to cues for the behavior	6. Environmental changes
Social influences	Social comparison Social support or encouragement (general) Information about others’ approval Social support (emotional) Social support (practical) Vicarious reinforcement Restructuring the social environment Modeling or demonstrating the behavior Identification of self as role model Social reward	11. Social processes of encouragement, pressure, support 12. Modeling/demonstration of behavior by others
Emotion	Reduce negative emotions Emotional consequences Self-assessment of affective consequences Social support (emotional)	5. Stress management 6. Coping skills
Behavioral regulation	Self-monitoring of behavior	2. Goal/target specified: behavior or outcome 3. Contract 4. Planning, implementation 5. Prompts, triggers, cues 6. Use of imagery

## Data Availability

Not applicable.

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
