# Peer review of "Exploring the Determinants of Polypharmacy Prescribing and Dispensing Behaviors in Primary Care for the Elderly—Qualitative Study"

_ijerph, 2023, doi:10.3390/ijerph20021389_

Round 1
Reviewer 1 Report
I appreciate the opportunity to review this manuscript on the important topic of polypharmacy in older adults. However, I have several concerns and would suggest major revision prior to re-considering publication of this study.
1. The authors focus on the term "polypharmacy" throughout the manuscript. However, over the last several years, there has been a move away from polypharmacy to consideration of appropriate and inappropriate prescribing in older adults. It would be important for the authors to clearly define what they are studying in the introduction. As is, they provide no definition for polypharmacy.
2. I found the presentation of methods a bit confusing and redundant. This could be helped with a visual representation of the steps of the study.
3. Inclusion of some of the comments from the interviews and focus group in the study would strengthen the manuscript and allow readers a betters sense of the input received from the GPs and pharmacist.
4. My most primary concern about this study is the decisions made by the researchers regarding what BCTs to include based, not on input from the interviews and focus group, but their own assessment of the literature and what would be/would not be feasible in the French medical system. The purpose of their qualitative approach is, per their introduction, to understand facilitators and barriers to GPs and pharmacists caring for the largest numbers of older adults in France. However, they then take it on themselves to change the conclusions from what is reported by their participants.
5. The authors do not present the limitations of their study. In addition, it is difficult to make any inferences regarding the generalizability of the study. It would be helpful to know more details about the types of primary care practices and pharmacies from which the participants were drawn (i.e. group practice versus single provider, academic or not, chain or independent pharmacy, etc.)
6. Moderate correction of English is needed before the manuscript would be acceptable for publication. Many of the sentences are poorly structured and, therefore, confusing. There are some misspellings and inappropriate punctuation. I would suggest review by a native English speaker.
Reviewer 2 Report
Notes:
1. The 1. table could be passed over; the names of municipalities and departments have low interest, the number of investigated areas should inclus into the text.
2. 4. and 5the tables are less informative, difficult to read, choose another structure for them
3. The French supplemantary materials unuseful for the majority of international readers, pass over !
Reviewer 3 Report
Dear authors
This is an interesting study about an important and current subject. The study is very interesting and is a further contribution to the area of polypharmacy management, in older adults, nevertheless are some points that needed to be clarified.
Comments:
The article is well written, well organized, easy to read and with a well-defined methodology.
However, the introduction and discussion should be improved with the inclusion/discussion of several papers on this theme published in regions with similar environments.
Regarding the methodology, it is the one usually used for this type of study, however it is important to understand why the number of participants is so low and why the focus groups do not have physicians and pharmacists together, could it not be more interesting?
Results:
Tables 4 and 5 should be reorganized, they are too extensive.
In 3.4. Selecting the components of the intervention (BCTs), the abbreviation should not be placed between () in the title, but in the text.
The legend of Figure 1 should be placed under the figure.
In section 5. Conclusions the abbreviations (BCTs) and (PTO) appear again, please review all these situations that appear throughout the text.
Round 2
Reviewer 1 Report
I appreciate you addressing the previously identified concerns. You have addressed them in a very complete manner and, at this time, I recommend publication of your manuscript.